# Economic burden of venous thromboembolism in surgical patients: A propensity score analysis from the national claims database in Vietnam

**My Hanh Bui[1,2]\*, Quang Cuong Le[3], Duc Hung Duong[4], Truong Son Nguyen[5], Binh Giang Tran[6], Tuan Duc Duong[7], Tien Hung Tran[7], Huu Chinh Nguyen[8], Thi Tuyet Mai Kieu[9], Hong Ha Nguyen[10], Long Hoang[11,12], Thanh Binh Nguyen[13], Thanh Viet Pham[14], Thi Hong Xuyen Hoang[2]**

1 Department of Tuberculosis and Lung Disease, Hanoi Medical University, Hanoi, Vietnam, 2 Department of Scientific Research and International Cooperation, Hanoi Medical University Hospital, Hanoi, Vietnam, 3 Department of Neurology, Hanoi Medical University, Hanoi, Vietnam, 4 Department of Cardiovascular Surgery, Bach Mai Hospital, Hanoi, Vietnam, 5 Director Board, Cho Ray Hospital, Ho Chi Minh City, Vietnam, 6 Director Board, Viet Duc Hospital, Hanoi, Vietnam, 7 Center for Health Insurance and Multilateral Payment in The Northern Region, Viet Nam Social Security, Hanoi, Vietnam, 8 Department of School and Occupational Nution, Nation Institute of Nutrition, Hanoi, Vietnam, 9 Department of Pharmaceutical Management and PharmacoEconomics, Hanoi University of Pharmacy, Hanoi, Vietnam, 10 Department of Maxillofacial, Plastic and Aesthetic Surgery, Viet Duc Hospital, Hanoi, Vietnam, 11 Department of Urology, Viet Duc Hospital, Hanoi, Vietnam, 12 Department of General surgery, Hospital of Hanoi Medical University, Hanoi, Vietnam, 13 Director Board, Hanoi Medical University Hospital, Hanoi, Vietnam, 14 Department of General Administration, Cho Ray Hospital, Ho Chi Minh City, Vietnam

\* buimyhanh@hmu.edu.vn

**Data Availability Statement:** The data are owned by a third party (Vietnam Social Security). The authors have to go to the Vietnam Social Security in order to access the data right there and and are

## Abstract

### Background

Venous thromboembolism (VTE) associated with surgery can cause serious comorbidities or death and imposes a substantial economic burden to society. The study examined VTE cases after surgery to determined how this condition imposed an economic burden on patients based on the national health insurance reimbursement database. Methods: This retrospective analysis adopted the public payer's perspective. The direct medical cost was estimated using data from the national claims database of Vietnam from Jan 1, 2017 to Sep 31, 2018. Adult patients who underwent surgeries were recruited for the study. Patients with a diagnostic code of up to 90 days after surgery were considered VTE cases with the outcome measure being the surgery-related costs within 90 days.

### Results

The 90-day cost of VTE patients was found to be US$2,939. The rate of readmission increased by 5.4 times, the rate of outpatient visits increased by 1.8 times and total costs over 90 days in patients with VTE undergoing surgery increased by 2.2 times. Estimation using propensity score matching method showed that an increase of US$1,019 in the 90-day cost of VTE patients.

supervised by a specialist of this agency. The authors do not have permission to share the national health insurance data. Requests for access should be directed to Vietnam Social Security: hungtran.it@gmail.com; (Tel: +84 45316681).

**Funding:** The authors received no specific funding for this work.

**Competing interests:** The authors have declared that no competing interests exist.

**Abbreviations:** ICD, International Classification of Diseases; PE, pulmonary embolism; PTP, phlebitis and thrombophlebitis; DVT, deep vein thrombosis; VET, venous embolism and thrombosis; VHIS, Vietnam Health Insurance Scheme; VTE, venous thromboembolism; RMPO, Readmission postoperative; OVP, Outpatient visit with problem.

## Conclusion

The VTE-related costs can be used to assess the potential economic benefit and cost-savings from prevention efforts.

## Introduction

Venous thromboembolism (VTE) is with the formation of blood clots that starts in the vein and may lead to long-term comorbidities or death [1], resulting in a significant economic burden on healthcare systems. With an estimated annual incidence of approximately 5–20 out of 10,000 persons [2–4], VTE is considered a common disorder. The occurrence of VTE in the Asian population has been lower than in others, but its incidence has been reported to have increased rapidly in the past decade [5, 6]. In Japan, postoperative patients without chemical thromboprophylaxis had a reported VTE incidence of 7.7% [7]. According to another study, out of 173 patients who had large open laparoscopic surgery, up to 24.3% were VTE cases [8]. A research in knee arthroplasty operation Asian patients with and without pharmacological prophylaxis suggested that who were administered chemoprophylaxis did not have a statistically significant difference in incidence of VTE although it may indicate better function [9]. A recent multicenter, observational, cohort involved 2,790,027 postoperative patients in Vietnam and showed that VTE was found in 3,068 patients (11 persons per 10,000) [10].

In addition to exposing patients to serious risks after major surgery [1], VTE also requires large healthcare expenditures due to its recurrence and complications [11]. A review of US database analyses indicated that costs of the initial VTE were approximately US$3,000–9,500. The total costs related to VTE over 3 months, 6 months, and 12 months were US$5,000, US$10,000 and US$33,000, respectively. Studies in the European Union countries showed lower additional inpatient costs after VTE (€1,800 after 3 months and €3,200 after one year); however it is still considered to have a substantial impact on healthcare systems. Complications related to VTE may require additional cost for treating the post-thrombotic syndrome (ranging from US$426 to US$11,700) and heparin-induced thrombocytopenia (ranging from US$3,118 to US$41,133) [11]. The development of a VTE event resulted in an increase in the length of hospitalization from 1.5-fold longer in patients undergoing major abdominal surgery [12] to a two-fold longer in patients after major orthopedic surgery [13]while the total costs after the surgery increased from 2 to 3.4-fold [12, 13].

Unlike other developed countries, Vietnam currently has no document describing the actual costs related to VTE in patients after surgery [11, 14]. Therefore, this study aims to investigate the economic burden of having a VTE event after surgery, using a national health insurance reimbursement database.

## Methods

### Data source

The database of the Vietnam Health Insurance Scheme (VHIS) was used in the study. The database contains health insurance claims for about 84% of the total population since 2017 [15]. Patient information including demography characteristics, ICD-10 code, medications, tests, surgeries, and diagnosis procedure combination claims is available in this database. Data were extracted on all in the time frame between Jan 1st 2017 and Sep 31st 2018.

## Study population

The surgical categories were included neurological, cardiothoracic, vascular, gastrointestinal, urologic, orthopedic, and plastic surgery. From the VHIS database, all adult patients ($\geq$18 years old) who experienced any of the above surgeries were recruited into the study. Patients were included if they underwent any of the following conditions: (1) diagnosis of VTE at the time of hospital admission and; (2) any anticoagulant treatment during the admission. Patients were excluded from the study if they underwent any of the following conditions: (1) pregnancy and; (2) contraindication to therapy of anticoagulant for any reason

## VTE cases

VTE cases were grouped into pulmonary embolism (PE) (using receipt codes equivalent to the International Classification of Diseases, Tenth Revision–ICD-10- code I26.0-I26.9), phlebitis and thrombophlebitis (PTP) (ICD-10 code equivalence: I80.1-I80.3; I80.8), and venous embolism and thrombosis (VET) (ICD-10 code equivalence I82). After the patient has been identified with a VTE code, cases with a diagnostic code up to 90 days after the first surgery will be considered potential cases.

## Matched controls

Patients in the cohort were 1: 1 case-control matched for a propensity score using a nearest-neighbor matching method. The propensity score was obtained by conducting a probit regression on all patients eligible for matching. The dependent variable was equal to 0 if the patient had no VTE and 1 if the patient had VTE. The independent variables included gender (male and female), age group (18–59, 60–69, 70–79, 80 and above), surgery type (neurological, cardiothoracic, vascular, gastrointestinal, urologic, orthopedic and plastic) and provider's area. Moreover, we also used 30 chronic conditions derived from the chronic conditions developed by Elixhauser et al. (Table 1) to adjust risk and match VTE cases [16]. To use the matching method, all continuous variables were classified categorically. All were eligible for inclusion and matched were included in the analysis.

## Cost analyses

The analysis used a linear regression method on a suitable sample to evaluate the link between VTE and the natural logarithm of the 90-day cost. The search was conducted from the payer's perspective, with outcome measure being the surgical-related costs during that period. All cost for local salaries are adjusted to control the difference in health costs corresponding to the region. Since the regression is implemented, the log of costs is then converted into dollars using the Duan's smearing estimator to adjust for the bias arising under the log retransformation [17]. To estimate the"excess total costs" due to VTE, we anticipate payment in two separate cases: all surgeries are "Yes" and "Not" including VTE and the variation between the two payments is the excess payment towing to VTE. Costs were presented in US dollar (US$). The exchange rate was calculated as US$1 = VND23,255 (2019).

## Ethics approval and consent to participate

This research was approved by the Ministry of Science and Technology in accordance with Decision No. 3622 and approved by the Ethics Committee of the Health Ministry No. 67 under the registry model. All patients agree verbally and/or in the pre-surgical written consent, all consent to the use of medical data and information for the hospital's training and research.

**Table 1. Characteristics of surgery patients.**

| | | No VTE | | VTE | Frequency (%) |
|---|---|---|---|---|---|
| | | **RAW sample** | **MATCHED sample** | | |
| | | **(n = 815,006)** | **(n = 1,612)** | **(n = 1,612)** | |
| Age group | 18–59 | 633,648 (77.75) | 681 (42.25) | 694 (43.11) | 0.11 |
| | 60–69 | 104,088 (12.77) | 375 (23.33) | 398 (24.69) | 0.36 |
| | 70–79 | 46,538 (5.71) | 261 (16.19) | 242 (15.07) | 0.56 |
| | > 80 | 30,732 (3.77) | 295 (18.24) | 278 (17.12) | 0.95 |
| Age (mean ± sd) | | 45.7 ± 17.3 | 60.5 ± 18.8 | 61.1 ± 17.5 | |
| Gender | Male | 487,581 (59.83) | 770 (47.83) | 784 (48.64) | 0.16 |
| | Female | 327,425 (40.17) | 842 (52.17) | 828 (51.36) | 0.26 |
| Region | North Region | 111,895 (13.73) | 134 (8.31) | 141 (8.81) | 0.12 |
| | Red River Delta | 186,863 (22.93) | 221 (13.77) | 200 (12.41) | 0.12 |
| | Central Coast | 209,209 (25.67) | 478 (29.65) | 500 (31.08) | 0.23 |
| | Central Highlands | 49,524 (6.08) | 71 (4.47) | 78 (4.84) | 0.14 |
| | Southeastern Region | 143,792 (17.64) | 400 (24.81) | 377 (23.45) | 0.28 |
| | Southwestern Region | 113,723 (13.95) | 308 (18.98) | 316 (19.42) | 0.27 |
| Surgery type | Neurosurgery | 51,202 (6.28) | 107 (6.64) | 75 (4.71) | 0.21 |
| | Cardiac-thoracic surgery | 13,815 (1.7) | 54 (3.41) | 56 (3.54) | 0.39 |
| | Vascular surgery | 5,806 (0.71) | 316 (19.6) | 392 (24.32) | 5.44 |
| | Gastrointestinal surgery | 273,449 (33.55) | 358 (22.21) | 303 (18.86) | 0.13 |
| | Urologic surgery | 18,911 (2.32) | 88 (5.52) | 88 (5.46) | 0.47 |
| | Orthopedic surgery | 425,425 (52.2) | 643 (39.89) | 654 (40.63) | 0.15 |
| | Plastic surgery | 26,398 (3.24) | 46 (2.73) | 44 (2.48) | 0.17 |
| Emergency | | 158,589 (19.46) | 345 (21.4) | 352 (21.9) | 0.22 |
| Chronic condition | Heart failure | 5,036 (0.62) | 61 (3.85) | 82 (5.15) | 1.63 |
| | Peripheral vascular disease | 1,540 (0.19) | 162 (10.05) | 168 (10.48) | 10.91 |
| | Paralysis | 807 (0.1) | 2 (0.19) | 10 (0.62) | 1.24 |
| | Rheumatoid arthritis | 8,230 (1.01) | 44 (2.73) | 42 (2.61) | 0.51 |
| | Gastric ulcer | 68,108 (8.36) | 282 (17.49) | 278 (17.25) | 0.41 |
| | Diabetes | 24,959 (3.06) | 223 (13.9) | 211 (13.15) | 0.85 |
| | Diabetes complications | 407 (0.05) | 4 (0.31) | 4 (0.31) | 0.98 |
| | Cancer | 36,213 (4.44) | 86 (5.4) | 76 (4.78) | 0.21 |
| | Metastatic cancer | 2,678 (0.33) | 8 (0.56) | 6 (0.43) | 0.22 |
| | Liver disease | 16,486 (2.02) | 90 (5.65) | 73 (4.53) | 0.44 |
| | Liver failure | 718 (0.09) | 2 (0.19) | 2 (0.12) | 0.28 |
| | Renal failure | 3,798 (0.47) | 94 (5.89) | 94 (5.83) | 2.47 |
| | Dementia | 152 (0.02) | 0 (0) | 0 (0) | 0.00 |
| | Alcohol abuse | 1,182 (0.15) | 0 (0.06) | 2 (0.19) | 0.17 |
| | Drug abuse | 44 (0.01) | 0 (0) | 0 (0.06) | 0.00 |
| | Deficiency anemia | 2,156 (0.26) | 18 (1.12) | 29 (1.8) | 1.35 |
| | Weight loss | 5,783 (0.71) | 16 (1.05) | 21 (1.3) | 0.36 |
| | Electrolyte disorders | 974 (0.12) | 16 (1.05) | 16 (1.05) | 1.64 |
| | Lymphoma | 1,227 (0.15) | 0 (0.06) | 2 (0.12) | 0.16 |
| | Hypothyroidism | 722 (0.09) | 8 (0.5) | 8 (0.5) | 1.11 |
| | Depression | 259 (0.03) | 0 (0) | 0 (0) | 0.00 |
| | Aplastic anemia | 2,530 (0.31) | 21 (1.3) | 23 (1.49) | 0.91 |
| | Arrhythmia | 4,253 (0.52) | 52 (3.23) | 52 (3.29) | 1.22 |
| | Valvular disease | 3,146 (0.39) | 29 (1.86) | 25 (1.55) | 0.79 |
| | Pulmonary vascular disease | 434 (0.05) | 2 (0.12) | 2 (0.12) | 0.46 |
| | Hypertension | 59,926 (7.35) | 468 (29.09) | 445 (27.61) | 0.74 |
| | Hypertension complications | 1,173 (0.14) | 14 (0.93) | 10 (0.62) | 0.85 |
| | Coagulopathy | 382 (0.05) | 0 (0) | 0 (0) | 0.00 |
| | Chronic pulmonary | 9,995 (1.23) | 73 (4.53) | 78 (4.84) | 0.78 |
| | Cerebral circulatory disease | 18,480 (2.27) | 132 (8.25) | 130 (8.13) | 0.70 |

VTE: Venous thromboembolism

RMPO: Readmission postoperative

OVP: Outpatient visit with problem

All patients' information is anonymous and data collection and analysis are performed by many people

## Statistical analyses

All analyses were conducted using Stata version 14.2 MP. A descriptive analysis was first performed to assess the characteristics of patients with and without complications. Categorical data were reported as absolute number (n) and proportion (%), and continuous data as mean with standard deviation (SD).

# Results

## Patient characteristics

During the study period, 1,612 (0,20%) of the 815,006 adult surgeries were identified as VTE cases. The VTE group had higher rate in women and mean age than the non-VTE group (52.17% vs. 47.83% and 61.1 year-old vs. 45.7 year-old), as shown in Table 1. We can see in the table that comorbidity rate in VTE surgeries is significantly higher than that of non-VTE surgeries in the raw sample. Therefore, the differences in individual characteristics may be the cause of the differences in costs and is presented in Table 2.

## Unadjusted cost and outcome

Table 2 presents the unadjusted cost and outcome of VTE and non-VTE group. The average 90-day surgery-related costs were $2,939 ± $3,834 for patients with VTE and $1,308 ± $1,923 for without VTE. The VTE patients who had come back to the hospitals for outpatient visits were 75.0% while this rate in non-VTE patients was 47.64%. Therefore, the cost of 90-day surgery-related outpatient visit of VTE patients was approximately 1.5-fold higher than the cost of 90 day surgery-related visits for non-VTE patients ($217.4 vs. $142.5). Notably, the readmission rate of VTE patients reached 43.92% which was rather high compared with 8.0% of non-VTE and the former's drug costs was three-time as much as the latter's ($455 vs. $141).

## Adjusted costs and outcomes

In order to reduce differences caused by variety in patient characteristics (Table 1), we present logistic regression estimates of odds ratios for patient outcomes after VTE events, controlling for patient characteristics after matching the 1,689 VTE cases to 1,689 non-VTE surgeries with similar characteristics.

## Readmissions due to VTE

In Table 3, PE was the category with the highest adjusted odds ratios for 90-day readmission during the 90-day postoperative period. PE had 10-fold larger odds of re-hospitalization, whereas VET and PTP had approximately 4-fold larger odds of readmission. The adjusted odds ratio of DVT+PE was 4.74. In summary, four of the VTE categories showed statistically significant positive acceleration in re-hospitalization rates as a result of VTE, from 23.9% to 49.7%. In Table 4, we show that the increased re-hospitalization rate caused by PE was estimated to be 51.7% (SE = 1%). The accelerated re-hospitalization rate was 28.8% (SE = 0.5%) for VET, 26.1% (SE = 0.3%) for PTP, and 38.9% (SE = 3.6%) for DVT+PE.

**Table 2. 90-day expenditures (raw) and outcome for surgery patients.**

| | No VTE | | VTE |
| --- | --- | --- | --- |
| | **RAW Sample** | **MATCHED Sample** | |
| | **(n = 815,006)** | **(n = 1,612)** | **(n = 1,612)** |
| RMPO rate (%) | 64,958 (7.97) | 217 (13.46) | 708 (43.92) |
| OVP rate (%) | 339,076 (41.6) | 768 (47.64) | 1209 (75) |
| Surgery payment ± SD (US$) | 524.2 ± 749.6 | 914.2 ± 1223.3 | 917.8 ± 1498.1 |
| Readmission payments ± SD (US$) | 600.6 ± 905.3 | 1100.3 ± 1394.6 | 1348.8 ± 1813.2 |
| RMPO drug payments ± SD (US$) | 122.2 ± 360.9 | 209 ± 434.8 | 358.1 ± 616.6 |
| Outpatient visit payments ± SD (US$) | 41.5 ± 216.4 | 142.5 ± 526.7 | 217.4 ± 652.5 |
| OVP drug payments ± SD (US$) | 19.2 ± 164 | 67.7 ± 270.4 | 97 ± 252 |
| Mean 90-day cost ± SD (US$) | 1,307.8 ± 1,922.5 | 2433.6 ± 2942.1 | 2939.1 ± 3834.3 |

VTE: Venous thromboembolism

RMPO: Readmission postoperative

OVP: Outpatient visit with problem

### Outpatient visits due to VTE

Table 3 indicates that PE had 1.79-fold larger rate for a 90-day outpatient visit with problems, PTP had a 3.9-fold larger rate of outpatient visit with problems, and VET and DVT+PE had 2.95-fold and 1.83-fold rate increase in outpatient visits with problems, respectively. The excess outpatient visit rate was 14.5% (SE = 1.5%) for PE, 25.4% (SE = 0.6%) for VET, 28.6% (SE = 0.5%) for PTP and 8.4% (SE = 5.3%) for DVT+PE. The increases in the outpatient visit rate were statistically significant in the four VTE categories (Table 4).

### Expenditure due to VTE

The overall costs associated with surgery through 90 days after surgery are showed in Fig 1. There was a separation of costs between patients with VTE and without VTE. The mean difference in costs markly advanced to US$499 after the procedure 30 days. The cumulative mean adjusted costs through 90 days reached the difference of US$632 between the VTE patient and control group. The difference demonstrated the economic impact of VTE on the healthcare system.

In Table 4, we simulate the increased 90-day costs caused by VTE from log-linear regression estimates for post-VTE payment, controlling for patient characteristics after pairing patient characteristics. The overall adjusted additional costs for the four VTE classes fluctuated from US$347 to US$2,651. In fact, the DVT+PE had the most considerable adapted additional post-discharge costs ($2651) in Table 4. PE leads to the second highest additional post-discharge cost of US$2,447. Readmission has the largest of the additional costs, from US$150 to US$1,182, followed by RMPO drug costs and outpatient visit costs.

### Discussion

Based on all the data collected, this is the first study describing the costs associated with VTE after surgery in Vietnam. The 90-day costs of VTE patients were found to be US$2,939.1. The rate of readmission increased by 5.5 times, the rate of outpatient visits increased by 1.8 times, and the total costs over 90 days in patients with VTE undergoing surgery increased by 2.25 times. Estimation using the propensity score matching method showed an increase of US $1,019 in 90-day costs for VTE patients. These results are later compared to those of previous

**Table 3. Estimated odds ratios for outcomes during the 90-day postoperative period.**

|  |  | RMPO | OVP |
|---|---|---|---|
| VTE | PE | 10.07* | 1.789* |
|  | PTP | 3.80* | 3.92* |
|  | VET | 4.32* | 2.95* |
|  | DVT+PE | 4.74 | 1.83 |
| Age group | 18–59 | - | - |
|  | 60–69 | 1.23 | 1.00 |
|  | 70–79 | 1.24 | 0.84 |
|  | > 80 | 1.09 | 0.53* |
| Gender | Male | - | - |
|  | Female | 0.88 | 1.18* |
| Area | North Region | - | - |
|  | Red River Delta | 1.06 | 1.07 |
|  | Central Coast | 0.93 | 1.12 |
|  | Central Highlands | 0.73 | 1.49 |
|  | Southeastern region | 0.65* | 2.88* |
|  | Southwestern region | 1.22 | 2.21* |
| Surgery type | Neurosurgery | - | - |
|  | Cardiac-thoracic surgery | 0.60 | 1.23 |
|  | Vascular surgery | 0.50* | 1.23 |
|  | Gastrointestinal surgery | 0.58* | 0.69 |
|  | Urologic surgery | 0.82 | 0.91 |
|  | Orthopedic surgery | 0.61* | 0.82 |
|  | Plastic surgery | 0.28* | 0.64 |

*p<0.05

VTE: Venous thromboembolism

RMPO: Readmission postoperative

OVP: Outpatient visit with problem

PTP: Phlebitis and thrombophlebitis

VET: Venous embolism and thrombosis

DVT: Deep Vein Thrombosis

studies; they overlap with each other with regard to the economic burden due to VTE's health-care system [11, 18]. However, VTE-related medical costs in Vietnam are rather small compared with developed countries. The results of US analyses indicated that VTE-related total costs over 6 months [19] reached US$10,000. In Japan, the 90-day costs after major abdominal surgery costs were US$20,648 [12]. Moreover, studies conducted in the EU indicated additional inpatient costs after VTE of €3,900 after 6 months [20]. The difference in health care systems and medical practices, as well as the affordability of VHIS, may account for the lower economic burden of VTE in Vietnam.

The incidence of postoperative VTE in this study was less than that of US hospitals (4.5 or more per 1,000 patients in 2010) [21, 22], but it had a similar incidence to that in Australian hospitals (2.5 vs. 2/1000 patients) [4]. The discrepancies and coding practices between our study and the one for Australia (ICD-10-AM) and the one for the USA (ICD-9-CM) may have contributed to this difference. It was shown that the accuracy of VTE coding can be improved by the adoption of extended codes developed in the revised ICD-9-CM. Our result is higher than that in a previous study implemented at four hospitals in Vietnam, in which VTE was

**Table 4.  Estimated increased 90-day outcomes and payments (adjusted) due to VTE (US$, 2019).**

| VTE class | Excess RMPO rates | Excess OVP rates | Excess Total cost | Excess RMPO cost | Excess RMPO Drug cost | Excess OVP cost | Excess OVP Drug cost | Excess hospital stays |
|---|---|---|---|---|---|---|---|---|
| All VTE | 0.296 | 0.246 | 1018.9 | 187.0 | 408.4 | 116.9 | 306.5 | 1.9 |
| | (0.003) | (0.003) | (18.567) | (4.996) | (7.553) | (2.031) | (4.089) | (0.042) |
| PE | 0.497 | 0.091 | 3182.0 | 825.8 | 1432.6 | 77.4 | 846.1 | 0.8 |
| | (0.012) | (0.012) | (106.735) | (16.741) | (29.934) | (11.851) | (25.006) | (0.075) |
| PTP | 0.292 | 0.268 | 770.2 | 93.9 | 312.1 | 113.2 | 251.1 | |
| | (0.004) | (0.003) | (20.368) | (3.538) | (6.298) | (3.01) | (3.982) | |
| VET | 0.338 | 0.249 | 1489.8 | 312.5 | 533.0 | 180.2 | 464.1 | 1.2 |
| | (0.006) | (0.006)(0.005) | (2.437)(30.481) | (5.715) | (2.485)(10.718) | (2.297)(4.521) | (2.768)(7.131) | (0.013) |
| DVT +PE | 0.389 | 0.084 | 2651.0 | 1182.3 | 588.5 | 160.2 | 287.3 | 0.5 |
| | (0.036) | (0.053) | (15.149) | (30.815) | (16.682) | (19.963) | (58.477) | (0.487) |

VTE: Venous thromboembolism

RMPO: Readmission postoperative

OVP: Outpatient visit with problem

PTP: Phlebitis and thrombophlebitis

VET: Venous embolism and thrombosis

DVT: Deep Vein Thrombosis

found in 3,068 patients among 2.8 million surgical cases (0.11%) [10]. This suggests that the use of systematic data result in an improved ability to monitor patients. We can detect VTE in postoperative patients even when they switch to treatment at other facilities.

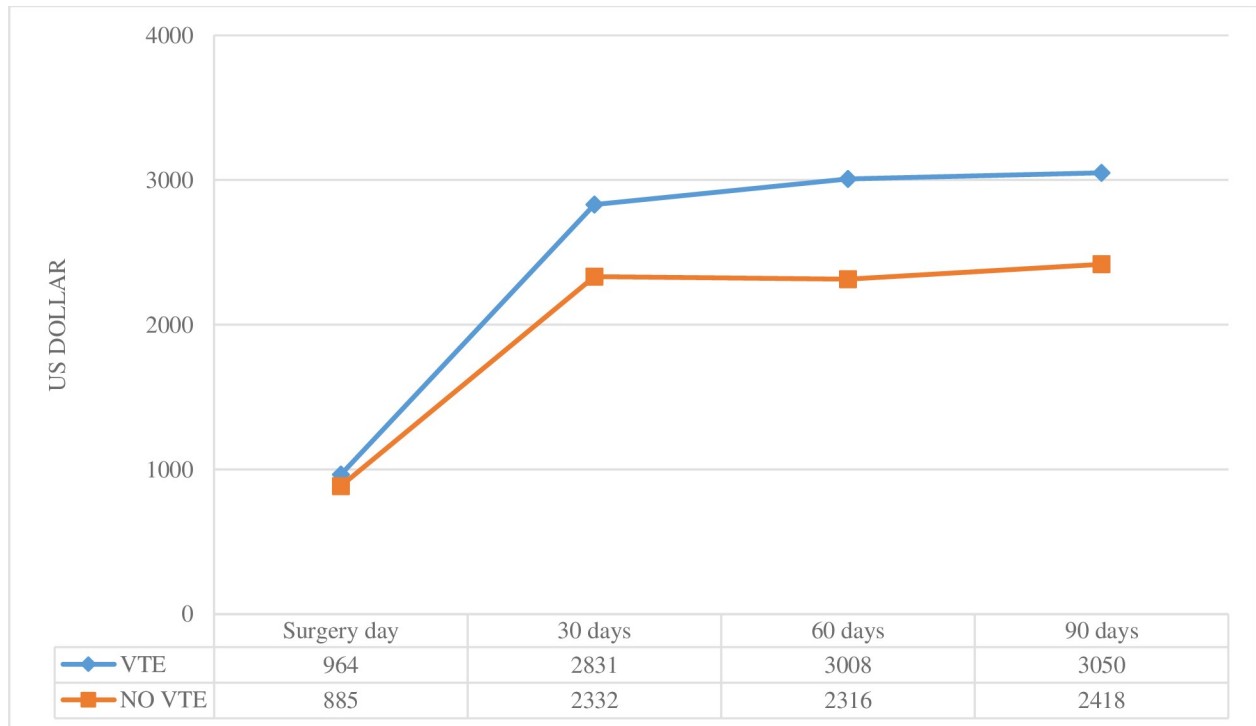

| | Surgery day | 30 days | 60 days | 90 days |
|---|---|---|---|---|
| VTE | 964 | 2831 | 3008 | 3050 |
| NO VTE | 885 | 2332 | 2316 | 2418 |

**Fig 1.  Cumulative mean adjusted costs from surgery day through 90 days after surgery in patients with venous thromboembolism and matched control patients.**

Aging was previously accepted as a main contributing factors with regard to the increasing trends in VTE rates for admitted patients in hospitals. Our finding that the incidence of VTE increases with advancing age, is similar to that observed by others [10, 23, 24]. We found that men were less likely to develop VTE complications, similar to the finding from Australian study [4]. Orthopedic, urologic and gastrointestinal surgery had the VTE incidence of 0.16%, 0.16% and 0.49%, respectively. The incidence is lower than that found in previous studies [8, 13] which only focus on major surgery. Further research should be conducted to examine the contributing factors for such a difference among different surgical procedures.

VTE remains a preventable complication. The large variance in VTE rates among different areas indicates that some hospitals are endeavoring to prevent the occurence of VTE. The national agency needs to develop a systematic program based on relevant experience in successfully reducing the VTE rate for difficult areas. The Vietnam National Society of Cardiology had developed recommendations, in which VTE prevention practices were promoted and related incidents were evaluated. Thus, national policies and local programs should focus on increasing the effectiveness of implementation the recommendations.

These are the first published estimates of incidences and costs of postoperative VTE in Vietnam using the national administrative database. This study provides economic evidence with regard to the need for stronger secondary prevention as a potential cost-saving approach. VTE is a preventable disease; therefore Ministry of Health should recommend using pharmacological prophylaxis to reduce both the clinical and economic burden in Vietnam.

This study has several limitations. First, only the ICD-10 code defined diagnosis was used to identify VTE in this study. This criterion could exclude VTE cases that were not recorded by code; however this would not have led to any significant difference in the costs. We also underestimated the VTE incidence because of these restrictive criteria, which would constitute a lower bound. Second, the use of insurance reimbursement database in Vietnam is still limited. Although some progress was noted using the database of the healthcare record system in Vietnam, it is still very limited compared to that of developed countries. However, VTE-associated mortality was not covered in the database, resulting in difficulties in estimating the burden of VTE related death. Finally, age group, gender, and surgery type may be a potential cause of the excess cost.

The findings suggest that clinical physicians and pharmacists should have a firm understanding of not only the clinical but also economic impacts of VTE. Appropriate and cost-effective prophylaxis and venous thromboprophylaxis methods guarantee the safety of patients as well as healthcare resources.

## Conclusion

VTE associated with surgery places a substantial economic burden on society. Our findings provide key cost parameters for assessing the cost-effectiveness of alternative interventions to reduce VTE occurrence and guiding reimbursement policy.

## Author Contributions

**Conceptualization:** My Hanh Bui, Duc Hung Duong, Truong Son Nguyen, Binh Giang Tran, Tuan Duc Duong, Tien Hung Tran, Thi Tuyet Mai Kieu, Hong Ha Nguyen, Thanh Viet Pham.

**Data curation:** My Hanh Bui, Tuan Duc Duong, Huu Chinh Nguyen, Hong Ha Nguyen.

**Formal analysis:** My Hanh Bui, Tien Hung Tran, Huu Chinh Nguyen.

**Funding acquisition:** My Hanh Bui.

**Investigation:** My Hanh Bui, Long Hoang.

**Methodology:** My Hanh Bui, Thi Tuyet Mai Kieu, Hong Ha Nguyen, Thanh Binh Nguyen.

**Project administration:** My Hanh Bui.

**Resources:** My Hanh Bui.

**Software:** Tien Hung Tran, Huu Chinh Nguyen.

**Supervision:** My Hanh Bui, Truong Son Nguyen, Binh Giang Tran.

**Validation:** My Hanh Bui.

**Writing – original draft:** My Hanh Bui, Tien Hung Tran, Thi Tuyet Mai Kieu, Thi Hong Xuyen Hoang.

**Writing – review & editing:** My Hanh Bui, Quang Cuong Le, Duc Hung Duong, Truong Son Nguyen, Tuan Duc Duong, Thi Tuyet Mai Kieu, Thi Hong Xuyen Hoang.

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
