## [Decision Letter · Decision Letter 0]

10 Dec 2019

PONE-D-19-28967

Economic Burden of Venous Thromboembolism in Surgical Patients: A propensity score analysis from national claims database in Vietnam

PLOS ONE

Dear Ms Kieu,

Thank you for submitting your manuscript to PLOS ONE. After careful consideration, we feel that it has merit but does not fully meet PLOS ONE’s publication criteria as it currently stands. Therefore, we invite you to submit a revised version of the manuscript that addresses the points raised during the review process.

Your manuscript received quite a bit of interest, and  I am including the comments from four of the reviewers who agreed to review, and who provided useful information to you regarding this submission. Your manuscript needs extensive revisions and you should try to respond to reviewers' comments and to fix your paper accordingly.

We would appreciate receiving your revised manuscript by Jan 24 2020 11:59PM. To enhance the reproducibility of your results, we recommend that if applicable you deposit your laboratory protocols in protocols.io, where a protocol can be assigned its own identifier (DOI) such that it can be cited independently in the future. For instructions see: http://journals.plos.org/plosone/s/submission-guidelines#loc-laboratory-protocols

We look forward to receiving your revised manuscript.

Kind regards,

Prof. Raffaele Serra, M.D., Ph.D

Academic Editor

PLOS ONE

Journal Requirements:

1.

2. In ethics statement in the manuscript and in the online submission form, please provide additional information about the patient records used in your retrospective study. Specifically, please ensure that you have discussed whether all data were fully anonymized before you accessed them and/or whether the IRB or ethics committee waived the requirement for informed consent. If patients provided informed written consent to have data from their medical records used in research, please include this information.

https://www.sciencedirect.com/science/article/pii/S2212109915000254?via%3Dihub

https://www.ncbi.nlm.nih.gov/pmc/articles/PMC2613997/

In your revision ensure you cite all your sources (including your own works), and quote or rephrase any duplicated text outside the methods section. Further consideration is dependent on these concerns being addressed.

4.

We note that you have indicated that data from this study are available upon request. PLOS only allows data to be available upon request if there are legal or ethical restrictions on sharing data publicly. For information on unacceptable data access restrictions, please see http://journals.plos.org/plosone/s/data-availability#loc-unacceptable-data-access-restrictions.

Reviewers' comments:

Reviewer's Responses to Questions

**Comments to the Author**

1. Is the manuscript technically sound, and do the data support the conclusions?

Reviewer #1: No

Reviewer #2: Partly

Reviewer #3: Yes

Reviewer #4: Partly

2. Has the statistical analysis been performed appropriately and rigorously? 

Reviewer #1: Yes

Reviewer #2: Yes

Reviewer #3: Yes

Reviewer #4: I Don't Know

3. Have the authors made all data underlying the findings in their manuscript fully available?

Reviewer #1: Yes

Reviewer #2: Yes

Reviewer #3: Yes

Reviewer #4: Yes

4. Is the manuscript presented in an intelligible fashion and written in standard English?

Reviewer #1: Yes

Reviewer #2: Yes

Reviewer #3: No

Reviewer #4: No

5. Review Comments to the Author

Reviewer #1: I have the following comments about this paper:

1. My main difficulty is that the authors have combined portal vein thrombosis (PVT) with DVT, PE and thrombophlebitis. PVT is often associated with general surgical problems and is caused by intra-abdominal inflammatory conditions, intra-abdominal sepsis, splenectomy, etc and this state is clearly different than DVT and PE. PVT is not considered one of the forms of VTE, which is usually DVT and PE. The authors should remove PVT and PTP (phlebitis and thrombophlebitis) from the VTE category and talk about them separately in the abstract and conlcusions.

2. Why would the authors exclude any anticoagulant treatment during the hospital admission? I would assume that most of the patients they are looking at would have received some type of DVT prophylaxis, so I would think this would have excluded most patients?

Reviewer #2: These results (other than the actual cost data for your country) are well known. What is the generalizable "new" message for the literature?

You did not mention VTE prophylaxis. What percent of patients with VTE received prophylaxis? Did you control for that in your propensity model>

Reviewer #3: 1. General comments

This study used a Vietnamese national healthcare insurance database in order to evaluate the incidence and costs associated with venous thromboembolism (VTE) in post-surgical patients over an almost 2 year period.

The methods of this paper are strong. Specifically, the use of propensity score matching is one of the major strengths of the paper. Furthermore, the results are compelling, and of interest to the target audience. Nonetheless, the paper is poorly written and difficult to follow in its current form.

I believe that the manuscript will be worthy of publication once it is thoroughly revised.

2. Abstract:

a. No specific comments beyond that detailed below for each section.

3. Introduction:

a. The introduction section would benefit from some grammar and sytax editing, but otherwise I have no major concerns regarding the introduction section. I would recommend including one additional comment regarding the incidence of VTE in post-operative patients who do receive chemical thromboprophylaxis, in addition to the single statement about incidence in patients who do not receive chemical prophylaxis, since most hospitalized post-op patients do receive some form of thromboprophylaxis.

4. Methods:

a. In the ‘Matched Controls’ section, the authors refer to Table 2 when discussing the Elixhauser Comorbidity Index, however Table 2 is unrelated to this statement.

5. Results:

a. Table 2 needs to be explained more clearly, and also needs to be discussed in the main text of the article. Currently, the only reference to Table 2 (in the Methods section, as I described above) is inaccurate.

b. There are many acronyms used throughout the results section that are not explained at any point – RMPO, VET, OVP, PTP, etc

c. Swap the order of mean ages in VTE vs. non-VTE group [sentence should read: “VTE group had higher rate of women and mean age than non-VTE group (50.9% vs. 40.2% and 60.5 year-old vs 45.7 year-old), as shown in Table 1]

d. Surgery is spelled incorrectly in the first column of Table 1 and in the title for Table 2.

6. Discussion:

a. Similar to the introduction section, the discussion section requires significant revisions to grammar and syntax, but otherwise no major concerns.

7. Figures and Tables:

a. Include a legend for each table, which includes what the acronyms stand for.

b. Refer to Table 2 somewhere in the main text (and delete the incorrect reference to Table 2 in the Methods section)

8. Conclusion:

a. No comments

9. References:

a. No comments

Reviewer #4: In this paper, the authors want to understand the economic burden caused by Venous Thromboembolism (VET) in Vietnamese patients. They use a propensity score matching to identify a control group of patients with similar observable characteristic who did not suffer VET within 90 of hospital admission and compare them to a set of patients that did. Using this technique, they find that the cost of VTE within 90 days of the hospital visit was a little over $1,000.

In full disclosure, I am not a medical expert and so the specific medical mechanisms tested in the paper are very new to me. A medical researcher should take a much more careful look at the medical and public health claims they made. However, I have researched in Vietnam for a long time and have some expertise in the econometric techniques they are using to estimate their effects. Because I am outside the specific field of the researchers, there may be different norms in statistical presentation. This should be taken into account in in assessing my review.

I think there is a lot to admire in this paper. The problem is compelling and the authors have assembled an excellent data for their analysis. I also think the wide variety of outcome variables studied provides a thorough picture of the problem. However, there is a lot that needs improvement in this paper before publication. In particular, the paper does not meet current standard for the presentation of econometric analysis. While medical research may be slightly different, the economic conventions require a great deal more information to make sense of the results.

1. The matching strategy is not well described. What specific type of matching estimator was used (propensity score, genetic, strata, entropy balancing? Did the authors employ weights? Did the match one-to-one or use multiple nearest neighbors? In fact, the authors don’t even show the matching estimation results in a table or appendix for the review to see. It is very hard to adjudicate the results the authors present without this information.

2. The key problem with matching strategies of all types is that one can only match on observables, when unobserved factors can drive both the presence of VET and the economic outcomes. The authors should make clear in their analysis what the potential threat from unobservable and what the potential direction of this bias is. In particular, I was concerned about whether VET is more likely in patients with particular baseline health profiles that are not picked up by the blunt covariates used in the match.

3. Table 3 is not well described enough to understand what is going on. Is this a logit analysis based on the matched sample or between the VET sample and the RAW population? Are these the odd ratios generated for each covariate from a logit model. I cannot tell because the table does not provide the sample size? Why is only statistical significance given and not the standard errors? Finally no log-likelihood, chi-squared, or any measure of model fit is given in the table.

4. Most importantly, it is highly unorthodox to control for covariates that were already included in the matching equation again as covariates in the estimation. What is the justification for this decision? If the authors are comparing to the matched sample, the covariates should be orthogonal to treatment by design and therefore irrelevant as control variables. This can lead to bias in the second stage.

5. Table 4 is even more confusingly described. Are those standard errors parentheses? Are the authors comparing to the matched sample again? Again, we can’t tell because no n is provided. Why are different covariates selected from those used in Table 3? What justifies this choice? Why are no estimates of model fit presented?

6. This statement is clearly biased upward by selection bias. It should not be the main finding presented in the abstract and conclusion. The matched estimates are half the size. “The 90-day cost of VTE patients was found to be US$2,913. There was a 5.4-fold increase in the rate of re-admission, 1.8-fold increase in the rate of outpatient visits and a 2.2- fold increase in total costs over 90 days in patients with VTE undergoing surgery.” The authors can put that in a footnote, but only the matched estimates should be used in presenting the size of the effects. They are the scientifically valid result.

7. I found the comparison to other countries in the discussion to be incomplete. Rather than providing the unadjusted estimates of VET costs for the US, Japan, and Europe, wouldn’t it make more sense to standardize by the cost of living in those countries (or at least by GDP per capita) to get a better sense of the true societal costs of the procedure.

8. Relatedly, to make sense of the costs of VET in the Vietnam context, it would be helpful to compare the cost of VET to the projected costs of prevention. This could be weighted by the probability of VET in each surgery type to get a sense of what the Vietnamese medical system should do to avoid the costs.

All in all, the authors have done important work. However more work on statistical presentation is necessary to adjudicate the findings and ultimately for the work to be understood and acted upon by a broad audience.

6. PLOS authors have the option to publish the peer review history of their article (what does this mean?). If published, this will include your full peer review and any attached files.

Reviewer #1: No

Reviewer #2: No

Reviewer #3: No

Reviewer #4: No

---

## [Author Response · Author response to Decision Letter 0]

3 Feb 2020

Thank you for your suggestion and contribution that help us better our research article

---

## [Decision Letter · Decision Letter 1]

4 Mar 2020

PONE-D-19-28967R1

Economic Burden of Venous Thromboembolism in Surgical Patients: A propensity score analysis from national claims database in Vietnam

PLOS ONE

Dear PhD Hanh,

Thank you for submitting your manuscript to PLOS ONE. After careful consideration, we feel that it has merit but does not fully meet PLOS ONE’s publication criteria as it currently stands. Therefore, we invite you to submit a revised version of the manuscript that addresses the points raised during the review process.

It os very important that you fully address all reviewers' concerns. If you want your article reconsidered, please pay particular attention to reviewer #3 comments.

We would appreciate receiving your revised manuscript by Apr 18 2020 11:59PM. To enhance the reproducibility of your results, we recommend that if applicable you deposit your laboratory protocols in protocols.io, where a protocol can be assigned its own identifier (DOI) such that it can be cited independently in the future. For instructions see: http://journals.plos.org/plosone/s/submission-guidelines#loc-laboratory-protocols

We look forward to receiving your revised manuscript.

Kind regards,

Prof. Raffaele Serra, M.D., Ph.D

Academic Editor

PLOS ONE

Additional Editor Comments (if provided):

There are still concerns about your revised manuscript, in particular see Reviewer's #3 comments.

Reviewers' comments:

Reviewer's Responses to Questions

**Comments to the Author**

1. If the authors have adequately addressed your comments raised in a previous round of review and you feel that this manuscript is now acceptable for publication, you may indicate that here to bypass the “Comments to the Author” section, enter your conflict of interest statement in the “Confidential to Editor” section, and submit your "Accept" recommendation.

Reviewer #1: All comments have been addressed

Reviewer #2: All comments have been addressed

Reviewer #3: (No Response)

2. Is the manuscript technically sound, and do the data support the conclusions?

Reviewer #1: Partly

Reviewer #2: Yes

Reviewer #3: Yes

3. Has the statistical analysis been performed appropriately and rigorously? 

Reviewer #1: I Don't Know

Reviewer #2: Yes

Reviewer #3: Yes

4. Have the authors made all data underlying the findings in their manuscript fully available?

Reviewer #1: Yes

Reviewer #2: Yes

Reviewer #3: Yes

5. Is the manuscript presented in an intelligible fashion and written in standard English?

Reviewer #1: Yes

Reviewer #2: Yes

Reviewer #3: No

6. Review Comments to the Author

Reviewer #1: I have no further comments for the authors, as my comments to the authors have been addressed successfully.

Reviewer #2: (No Response)

Reviewer #3: The authors did not address any of the comments that I made in my initial review. I had major concerns about the manuscript as it was initially submitted, and since the authors did not address any of those concerns, I am not recommending that the manuscript be rejected.

7. PLOS authors have the option to publish the peer review history of their article (what does this mean?). If published, this will include your full peer review and any attached files.

Reviewer #1: No

Reviewer #2: No

Reviewer #3: No

---

## [Decision Letter · Decision Letter 2]

24 Mar 2020

Economic Burden of Venous Thromboembolism in Surgical Patients: A propensity score analysis from national claims database in Vietnam

PONE-D-19-28967R2

Dear Dr. Hanh,

We are pleased to inform you that your manuscript has been judged scientifically suitable for publication and will be formally accepted for publication once it complies with all outstanding technical requirements.

With kind regards,

Prof. Raffaele Serra, M.D., Ph.D

Academic Editor

PLOS ONE

Additional Editor Comments (optional):

amended manuscript is acceptable.

Reviewers' comments:

Reviewer's Responses to Questions

**Comments to the Author**

1. If the authors have adequately addressed your comments raised in a previous round of review and you feel that this manuscript is now acceptable for publication, you may indicate that here to bypass the “Comments to the Author” section, enter your conflict of interest statement in the “Confidential to Editor” section, and submit your "Accept" recommendation.

Reviewer #1: All comments have been addressed

2. Is the manuscript technically sound, and do the data support the conclusions?

Reviewer #1: Yes

3. Has the statistical analysis been performed appropriately and rigorously? 

Reviewer #1: Yes

4. Have the authors made all data underlying the findings in their manuscript fully available?

Reviewer #1: Yes

5. Is the manuscript presented in an intelligible fashion and written in standard English?

Reviewer #1: Yes

6. Review Comments to the Author

Reviewer #1: I have no additional comments for the authors. I had already suggested that this manuscript be accepted.

7. PLOS authors have the option to publish the peer review history of their article (what does this mean?). If published, this will include your full peer review and any attached files.

Reviewer #1: No

---

## [Editor Report · Acceptance letter]

26 Mar 2020

PONE-D-19-28967R2 

 Economic Burden of Venous Thromboembolism in Surgical Patients: A propensity score analysis from the national claims database in Vietnam 

Dear Dr. Bui:

I am pleased to inform you that your manuscript has been deemed suitable for publication in PLOS ONE. Congratulations! Your manuscript is now with our production department. 

With kind regards,

on behalf of

Prof. Raffaele Serra 

Academic Editor

PLOS ONE